# Deep Learning for Human Activity Recognition on 3D Human Skeleton: Survey and Comparative Study

**DOI:** 10.3390/s23115121

**Published:** 2023-05-27

**Authors:** Hung-Cuong Nguyen, Thi-Hao Nguyen, Rafał Scherer, Van-Hung Le

**Affiliations:** 1Faculty of Engineering Technology, Hung Vuong University, Viet Tri City 35100, Vietnam; cuongnh@hvu.edu.vn (H.-C.N.); haont@hvu.edu.vn (T.-H.N.); 2Department of Intelligent Computer Systems, Czestochowa University of Technology, 42-218 Czestochowa, Poland; rafal.scherer@pcz.pl; 3Faculty of Basic Science, Tan Trao University, Tuyen Quang City 22000, Vietnam

**Keywords:** human activity recognition, 3D human pose/skeleton, deep neural networks, recurrent neural networks (RNN), convolutional neural networks (CNN), graph convolution networks (GCN), KLHA3D 102 dataset, KLYoga3D dataset

## Abstract

Human activity recognition (HAR) is an important research problem in computer vision. This problem is widely applied to building applications in human–machine interactions, monitoring, etc. Especially, HAR based on the human skeleton creates intuitive applications. Therefore, determining the current results of these studies is very important in selecting solutions and developing commercial products. In this paper, we perform a full survey on using deep learning to recognize human activity based on three-dimensional (3D) human skeleton data as input. Our research is based on four types of deep learning networks for activity recognition based on extracted feature vectors: Recurrent Neural Network (RNN) using extracted activity sequence features; Convolutional Neural Network (CNN) uses feature vectors extracted based on the projection of the skeleton into the image space; Graph Convolution Network (GCN) uses features extracted from the skeleton graph and the temporal–spatial function of the skeleton; Hybrid Deep Neural Network (Hybrid–DNN) uses many other types of features in combination. Our survey research is fully implemented from models, databases, metrics, and results from 2019 to March 2023, and they are presented in ascending order of time. In particular, we also carried out a comparative study on HAR based on a 3D human skeleton on the KLHA3D 102 and KLYOGA3D datasets. At the same time, we performed analysis and discussed the obtained results when applying CNN-based, GCN-based, and Hybrid–DNN-based deep learning networks.

## 1. Introduction

HAR is an important research problem in computer vision. It is applied in many fields, such as human–machine interaction [1], video surveillance [2,3], and fashion and retail [4]. It has been of research interest for nearly a decade, and studies are often based on human pose to perform the activity recognition process, where especially 3D human pose-based activity recognition provides real-world-like visualization.

Despite much research interest and impressive results, HAR still contains many real challenges in the implementation process. In Islam et al. [5]’s study, the following challenges were presented. Firstly, the skeleton data of commonly seen daily activities such as walking, running, and sitting down are often recognizable and rich in data. However, data on the skeleton of concurrent or aggregate actions on these actions are sorely lacking. Secondly, the datasets of 3D human skeletons currently used to train recognition models are often in a scene with only one person (a skeleton); in real-life operations, there are often many people performing many activities, such as queuing in a store, walking, or jogging. In particular, information about the context of people’s activities is still lacking. Since human activity recognition is very closely related to understanding human behavior, context plays a huge role in HAR. Thirdly, human skeleton datasets for HAR can be collected from many different types of sensors, different objects, and different contexts. Although they perform the same activity, some people are taller or shorter, the method of execution is not the same for each person, etc., and has not been standardized. Especially, 3D human skeleton data can be represented in spatial and temporal models; the number of dimensions of the data is too large because the data of the 3D human skeleton has many degrees of freedom, or due to the exploitation of different operations, movement is located on only a small part of the human body.

In the studies of Xing et al. [6], Ren et al. [7], and Arshad et al. [8], they conducted a full survey of HAR based on a 3D human skeleton, in which the research is also surveyed and divided into three approaches: RNN-based, CNN-based, and GCN-based. In these two survey studies, the challenges of HAR based on the 3D human skeleton have not been presented and analyzed. Islam et al. [5] only conducts the survey using CNN-based HAR.

In this paper, we conduct a survey on deep learning-based methods, datasets, and HAR results based on 3D human poses as input data. From there, we propose and analyze the challenges in recognizing human activities based on the 3D human pose.

Previous studies on HAR often applied to datasets with a small number of 3D joints, namely from 20 to 31 points as the HDM05 dataset [9] (31 3D joints), CMU dataset (22 3D joints) [10], NTU RGB-D dataset (25 3D joints) [11,12], MSRA Action3D dataset (20 3D joints) [13], as presented in Figure 1 of Kumar et al. [14]’s research. With a low number of representative points, the low dimensionality of the data is provided, but the lack of enough information to distinguish actions also occurs. As illustrated in Figure 1, the number of joints is large and has many degrees of freedom, which makes the dimensionality of the data very large, especially when the skeleton moves in the 3D space and follows the temporal model. Another problem is that in the KLHA3D-102 [14] dataset, there are many similar actions such as “drink water” and “drink tea” and “golf swing pitch shot” and “golf swing short shot”.

In the study of Kumar et al. [14], the problem of human activity recognition was researched on the HDM05, CMU, NTU RGB-D, MSRA Action3D, KLHA3D102 [14], and KLYOGA3D [14] datasets. At the same time, we also perform experiments on activity recognition on the DDNet, Res-GCN, and PA Res-GCN models with the KLHA3D102 [14], and KLYOGA3D [14] datasets. To scrutinize the results of HAR based on the latest deep learning models (DLM), we have performed a comparative study on HAR on the KLHA3D102 [14] and KLYOGA3D [14] datasets.

The main contributions of the paper are as follows:We present an overview of the HAR problem based on the 3D human pose as the input, with four types of DNN to perform the estimation: RNN-based, CNN-based, GCN-based, and Hybrid–DNN-based.A full survey of HAR based on the 3D human pose is elaborated in detail from methods, datasets, and recognition results. More specifically, our survey provided about 250 results of the HAR across more than 70 valuable studies from 2019 to March 2023. The results are listed in ascending order of the year and are all evaluated on the accuracy measure.We performed a study comparing HAR on the KLHA3D 102 [14] and KLYOGA3D [14] datasets with the GCN-based and Hybrid–DNN-based neural models.Analysis of challenges in HAR based on the 3D skeleton of the whole body is presented. The analysis of the challenges of implementing HAR with two main contents is the number of dimensions of the data and the insufficient information to distinguish actions with a limited number of reference points.

The content of this paper is organized as follows. Section 1 introduces the applications and difficulties of HAR based on 3D human skeleton input data. Section 2 discusses related research in HAR. Section 3 presents a full survey of HAR methods based on 3D human skeleton data input. Section 4 presents a comparative study of HAR on the KLHA3D 102 [14] and KLYOGA3D [14] datasets. Section 5 concludes the contributions and presents future works.

## 2. Related Works

HAR is based on computer vision with RGB, skeletal, and depth input representation. Wang et al. [15] surveyed on HAR based on input data can be the skeleton, RGB, RGB + D, optical flow, etc. In the study, the authors only present and analyze the ST-GCN (Spatial Temporal Graph Convolutional Networks) [16] and 3D CNNs (3D Convolutional Neural Networks) hybrid with some architecture [17,18]. HAR results were not presented in this study.

Morshed et al. [19] conducted a comprehensive survey of HAR with three methods of input data types: depth-based methods, skeleton-based methods, and hybrid feature-based methods. They showed that the approaches using depth information could use more RGB images to extract features such as Histogram of Gradients (HOG) to generate Depth Motion Maps (DMM) and to train HAR models. With skeleton-based methods, the trajectory-based method used is based on the trajectory that investigates the spatial and temporal movement of the human body’s skeleton to extract different features. The human skeleton can be a 2D skeleton or a 3D skeleton. At the same time, the volume-based methods usually compute features like texture, color, pose, histograms of optical flow, histograms of directed gradients, and other features to represent human activity in a spatial–temporal volume. The results in Tables 1 and 2 from [19] are mainly created on RGB and depth videos, with only one result [20] on the MSRAction3D dataset.

In addition, there is a survey by Jobanputra et al. [21] divided into two main directions: using traditional machine learning and deep learning. Within deep learning, they separated dense artificial neural networks, convolutional neural networks (CNN), and recurrent neural networks (RNN).

Gupta et al. [22] surveyed HAR of people from multi-modal information sources, where information can be sensor-based [23], vision-based [24], RFID-based [25,26], WiFi-based [27,28], and device free [29]. The recognition model is also based on two methods: machine learning and deep learning. They calculated the ratio of device types to capture the data for HAR or perform HAR right on those devices as follows: vision = 34%, WiFi = 7%, RFID = 7%, sensor = 52%, and the ratio of vision-based input information is video = 65% and skeleton = 35%, as illustrated in Figure 2.

## 3. HAR Based on 3D Human Pose: Survey

### 3.1. Methods

In the valuable surveys of Ren et al. [7] and Le et al. [30], the problem of HAR using 3D skeleton-based deep learning models can be solved by four types of deep learning models (DLMs): recurrent neural networks, convolutional neural networks, graph convolution network, and hybrid deep neural networks (Hybrid-DNN). The types of deep learning models and methods for HAR are illustrated in Figure 3. In Xing et al. [6] and Ren et al. [7], they conducted a full survey on HAR based on 3D human skeletons, however, the results have only been updated until 2019. In this paper, we update the results of HAR based on a 3D skeleton from 2019 to March 2023.

#### 3.1.1. RNN-Based

RNN-based methods that use vector sequences of joint positions in continuous time, the position of each joint in the human body as it moves over time can be expressed as a vector. The main idea of the RNN is to use some kind of memory to store information from previous computational steps so that based on it can make the most accurate prediction for the current prediction step. Figure 4 illustrates the RNN-based approach for HAR based on a 3D human skeleton. As known, a variation of RNN called Long Short-Term Memory (LSTM) [31,32] has achieved many impressive results in the field of NLP (Natural Language Processing) [33,34], speech recognition [35], and especially in computer vision [36,37]. LSTM is the same as a traditional RNN, except adding computational gates in the hidden layer to decide to retain important information for the next time steps.

Ye et al. [38] proposed a combination of RNN and LSTM to learn geometric features from 3D human skeleton data. The proposed model selects geometric features based on distances from joints and selected lines as the input of the network. To this end, the model has the first layer of LSTM and the second layer of temporal pooling with the ability to select the most recognizable time period features. To extract the geometric features, the algorithm performs two steps. The first is pre-processing the 3D human skeleton data by converting the 3D human skeleton data from the camera coordinate system to the human coordinate system, with the origin being the center of the hip joints. The X-axis is a 3D vector parallel to the “Right shoulder”| and the “Left shoulder” (red axis), the Y-axis is parallel to the 3D vector from the “Head” to the “Center hip”, and the Z-axis is then X × Y. The coordinate system on the body is shown in Figure 5. The second step is to represent the geometric features, unlike other studies that use the coordinates of the joints as the input. In this study, the authors use 30 lines selected on the lines as shown in Figure 5 as the input for geometric feature calculation.

Gaur et al. [37] develops a HAR framework based on LSTM-RNN. The framework is integrated into wearable sensors. The proposed framework includes four modules: the first is the data pre-processing, the second describes the benefits and drawbacks of the RNN model, the third is the LSTM networks model, and the final is the combination module of LSTM and RNN.

Li et al. [39] proposed an independently recurrent neural network (IndRNN) to solve five problems of RNN for HAR. The first one can process longer sequences greater than 5000 steps and still solve the problem of gradient vanishing and exploding. The second can construct deeper networks (over 20 layers, much deeper if GPU memory supports). The third can be robustly trained with ReLU. The fourth can be to interpret the behavior of IndRNN neurons independently without the effect of the others. The fifth is reduced computational complexity (over 10 times faster than cuDNN LSTM when the sequence is long).

Liao et al. [40,41] proposed the Logsig-RNN with some advantages over RNN as follows: (1) The sequence of the log signatures at a coarser time partition is transformed from a high-frequency sampled time series by the log-signature layer. The log-signatures transformation reduces training time. (2) When using high frequency and continuous data sampling at a coarser time grid, it is possible to ignore the microscopic character of the streamed data and render lower accuracy. Meanwhile, the Logsig-RNN model can do this well. (3) has a much better performance than RNN when performed on missing data. (4) The Logsig-RNN model can sample the highly oscillatory stream data. The improved model of Logsig-RNN compared to RNN is illustrated in Figure 6.

#### 3.1.2. CNN-Based

This approach utilizes the outstanding advantages of CNN in object recognition in 2D space (image space). It maps a 3D representation of the 3D human skeletons into a 2D array (possibly spatial relations from the skeleton joints) to learn the spatial-temporal skeleton features. The CNN-based approach is illustrated in Figure 7.

Tasnim et al. [42] proposed a DCNN model to train the feature vector transformed from the coordinates of the joints along the *X*, *Y*, and *Z* axes. The joints of each frame *i*th are represented by $F_{i}\left(X_{ij},Y_{ij},Z_{ij}\right)$, where *j* is the joint number. Li et al. [43] proposed a CNN fusion model for skeletal action recognition. The fusion model was trained from two types of feature vectors: three SPI (skeletal pose image) sequences and three STSIs (skeletal trajectory shape images). A PoseConv3D model was proposed by Duan et al. [44]. This model used the 3D-CNN for capturing the spatio-temporal dynamics of skeleton sequences; there in the input of the 3D-CNN backbone are 3D heatmap volumes. The pseudo heatmaps for joints and limbs are generated and are good inputs for 3D-CNNs. Koniusz et al. [45] propose the sequence compatibility kernel (SCK) and dynamics compatibility kernel (DCK) feature representations. SCK is generated from the spatio-temporal correlations between features as illustrated in Figure 2a,b of Koniusz et al. [45]’s research, and DCK explicitly models the action dynamics of a sequence as illustrated in Figure 4a,b of Koniusz et al. [45]’s research. This research used the ResNet-152 model [46] to train the HAR features.

#### 3.1.3. GCN-Based

GCN-based deep learning uses the natural representation of the 3D human skeleton as a graph, with each joint as a vertex and each segment connecting the human body parts as an edge. This approach often extracts the spatial and temporal features of the skeleton graph series, as illustrated in Figure 8.

With the advantages of features that can be extracted from the skeleton graph, this approach has received much research attention in the past four years. Figure 9 shows the number of studies based on the GCN methods.

In 2019, Shi et al. [47] proposed a novel MS-AAGCN (multistream attention-enhanced adaptive graph convolutional neural network) with some advantages for HAR based on 3D human skeleton data. The first is that an adaptive GCN is proposed to adaptively learn the topology of the graph in an end-to-end manner. The second is to embed an STC-attention module in each graph convolutional layer, which can help the model learn to focus on discriminative joints, frames, and channels selectively. Third, the combination of information from bones, joints, and information about the movement of bones and joints has created high efficiency for the activity recognition process. Peng et al. [48] proposed the first automatically designed GCN as well as a NAS (Neural Architecture Search). The spatial-temporal correlations between nodes are used to increase the search space of the GCN by building higher-order connections with a Chebyshev polynomial approximation. The NAS helps to increase search efficiency; it both performs sampling and is memory-efficient. Shi et al. [49] proposed a novel directed graph neural network to train features extracted from joints, bones, and their relationships. The skeleton data are represented as a DAG (directed acyclic graph) based on the kinematic dependency between the joints and bones in the human body. A two-stream framework is used to exploit two streams of information, namely the space and time of movement of the joints. The AS-GCN (actional-structural graph convolution network) is proposed by Li [50]. The AS-GCN combines both actional-structural graph convolution and temporal convolution into a basic building block for training both spatial and temporal features. The AS-SCN block is connected to two parallel branches by a future pose prediction head.

A novel end-to-end network AR-GCN (attention-enhanced recurrent graph convolutional network) is proposed by Ding et al. [51]. AR-GCN is an end-to-end network capable of selectively learning discriminative spatial-temporal features and overcoming the disadvantages of learning only using key frames and key joints. The AR-GCN combines the advantages of the GCN and an RNN. Thus, the AR-GCN promotes the spatial feature extraction ability of GCN and improves the discriminative temporal information modeling ability. Gao et al. [52] proposed the BAGCN (Bidirectional Attentive Graph Convolutional Network). A GCN-based focusing and diffusion mechanism is used to learn spatial-temporal context from human skeleton sequences. The features of BAGCN are built based on the representation of the skeleton data in a single frame by two opposite-direction graphs, thereby effectively promoting the way of message passes in the graph. Li et al. [53] proposed the Sym-GNN (Symbiotic Graph Neural Networks) for HAR and predicting motion based on a 3D human skeleton. Sym-GNN consists of two component networks: a prime joint-based network to learn body-joint-based features, and a dual bone-based network to learn body-bone-based features. The backbone of each network is essentially a multi-branch, multi-scale GCN. Wu et al. [54] proposed a dense connection block for ST-GCN to learn global information, and to improve the robustness of features. The proposed method based on the spatial residual layer and the dense connection block produces better results than state-of-art methods resting on the spatial-temporal GCN. A two-stream non-local graph convolutional network is proposed by Shi et al. [55] to solve the problem of mining both the coordinate of joints and the length and information direction of bones. The BP algorithm is used to learn the topology of the graph in each layer. Papadopoulos et al. [56] proposed the DH-TCN (Dilated Hierarchical Temporal Graph Convolutional Network) module for modeling short and long-term dependencies. To represent extracted features on a 3D human skeleton, the author proposed a GVFE (Graph Vertex Feature Encoder) module for encoding vertex features.

Kao et al. [57] proposed graph-based motion representations using the skeleton-based graph structure. A skeletal-temporal graph starts with a skeletal-temporal graph such as the Fourier transform graph. The skeletal-temporal graph is transformed into the motion representation. To extract features for the activity recognition process, the authors used temporal pyramid matching [58] to model the dynamics in the sequence of frame-wise representations.

In 2020, Song et al. [59] proposed the PA-ResGCN with the combination of MIBs (Multiple Input Branches), ResGCN (Residual GCN) with a bottleneck structure, and PA (Part-wise Attention) blocks. The authors calculated and characterized spatial-temporal sequence from the joints, velocity, and bone of the human skeleton based on human body parts. These features are represented by a part of the human skeleton and trained by some Residual GCN modules. Next, the branches are concatenated and sent to several PA-ResGCN modules, where each PA-ResGCN module contains a sequential execution of a Residual GCN module.

The Shift-GCN is proposed by Cheng et al. [60]. Other GCNs, such as AS-GCN and Adaptive GCN, use heavy regular graph convolutions. The Shift-GCN uses shift graph operations and lightweight point-wise convolutions. In the shift graph, both the spatial graph and temporal graph are used to compute feature vectors. Thus, the computational complexity is significantly reduced. Song et al. [61] proposed the GCN-based multi-stream model called the RA-GCN (richly activated GCN). The rich discriminative features are extracted from skeleton motion sequences. Especially, the noisy or incomplete skeleton data brings challenges to HAR and training; thus, RA-GCN proposed the problem with the learned redundant. Peng et al. [48] proposed a brand-new ST-GCN to model the graph sequences on the Riemann manifold by Poincare geometry features computed from the spatial-temporal graph convolutional network. A Poincare model is trained on a multidimensional structural embedding for each graph. Mixing the dimensions is used to provide a more distinguished representation of the Poincare model. To obtain effective feature learning, Liu et al. [62] proposed a unified spatial-temporal graph convolution called G3D. This method is based on the multi-scale aggregation scheme to remove the redundant dependencies between node features from different neighborhoods. G3D introduced graph edges across the “3D” spatial-temporal domain as skip connections for the unobstructed information flow. The Dynamic GCN proposed by Ye et al. [63] exploits the advantages of learning-based skeleton topology of CNNs. A CNN named CeN (Context-encoding Network) is introduced to learn skeleton topology automatically. CeN can be embedded into a graph convolutional layer and learned end-to-end. The contextual information of each joint can be monitored globally by CeN and can represent the dynamics of the skeleton system more accurately. Obinata et al. [64] proposed extending the temporal graph of a GCN. The authors performed adding connections to neighboring multiple vertices on the inter-frame and extracting additional features based on the extended temporal graph. From this, the extended method can extract correlated features of multiple joints in human movement for training the HAR model. Yang et al. [65] proposed a PGCN-TCA (pseudo graph convolutional network with temporal and channel-wise attention) to solve the three existing problems of the previous GCN-based networks. First, the features of joints are usually only extracted based on the direct connection between bones for which the distant joint information that has no physical connection in a skeleton chain has not been used. The second is the normalized adjacency matrices are directly computed by the predefined graph and kept fixed through the training process. They are used on most GCN-based networks, which makes the model unable to extract diverse features. The third is that different frames and channels are of different importance to action recognition.

Ding et al. [66] proposed a novel SemGCN (Semantics-Guided Graph Convolutional Network) to extract multiple semantic graphs for skeleton sequences adaptively. This method can explore action-specific latent dependencies, and allocate different levels of importance to different skeleton information. The spatial useful and temporal information is extracted based on the different feature fusion strategies of the Sem-GCN block.

Yu et al. [67] proposed PeGCNs (Predictively encoded Graph Convolutional Networks) to train a GCN-based action recognition model with missing and noisy human skeleton data. To learn such representations by predicting the perfect sample from the noisy sample in latent space via an auto-regression model by using a probabilistic contrastive loss to capture the most useful information for predicting a perfect sample.

The PR-GCN (pose refinement graph convolutional network) is proposed by Li et al. [68]. To reduce the impact of errors in the skeleton data, the authors preprocessed the input skeleton sequences via a pose refinement module. Then, the position and motion information is combined through two branches: a motion-flow branch and a position-flow-branch. In addition, the refined skeleton sequences are created based on gradual fusion. Finally, the temporal aggregation module aggregates the information over time and predicts the action class probabilities.

In 2021: Chen et al. [69] proposed a dual-head GNN (graph neural network) framework for HAR based on human skeleton data. This method used two branches of interleaved graph networks to extract features at two different temporal resolutions. The branch with a lower temporal resolution captures motion patterns at a coarse level, and the branch with a higher temporal resolution is encoded time movements on a more sophisticated level. These two branches are processed in parallel, and the output is the dual-granular action classification. Yang et al. [70] proposed a new framework called UNFGEF. This framework is unified with 15 graph embedding features with GCN and model characteristic skeletons. The human skeleton is represented using the adjacent matrix to represent the skeleton graph. The graph features of nodes, edges, and subgraphs are extracted and embedded into GCN and TCN networks. The final prediction is fused from the multi-stream through the softmax classifier for each stream. Chen et al. [71] propose the CTR-GC (Channel-wise Topology Refinement Graph Convolution) to learn different topologies dynamically and effectively aggregate joint features in different channels. This method learns a shared topology and channel-specific correlations simultaneously. To solve the problem of confusion between the activities of the nearly identical human bodies, Qin et al. [72] proposed fusing higher-order features in the form of angular encoding (AGE) into modern architectures to capture the relationships between joints and body parts robustly. To extract relevant information from neighboring nodes effectively while suppressing undesired noises, Zeng et al. [73] suggested a hop-aware hierarchical channels-squeezing fusion layer. The information from distant nodes is extracted and fused in a hierarchical structure. Dynamic skeletal graphs are built upon the fixed human skeleton topology and capture action-specific poses. Song et al. [74] have made improvements to the ResGCN to EfficientGCN v1, the authors used additional three types of layers (SepLayer, EpSepLayer, and SGLayer) for skeleton-based action recognition. This study employs a compound scaling strategy to configure the model’s width and depth with a scaling coefficient; since then, the number of hyper-parameters is also calculated automatically. EfficientGCN v1 considers spatial attention and distinguishes the most important temporal frames. To extract effective spatial-temporal features from skeleton data in a coarse-to-fine progressive process for action recognition, Yang et al. [75] suggested the FGCN (Feedback Graph Convolutional Network). FGCN builds a local network with lateral connections between two temporal stages by a dense connections-based FGCB (Feedback Graph Convolutional Block) to transmit high-level semantic features to low-level layers.

In 2022: Lee et al. [76] proposed the HD-GCN (hierarchically decomposed graph convolutional network), as illustrated in Figure 10. HD-GCN contains a hierarchically decomposed graph (HD-Graph) to thoroughly identify the distant edges in the same hierarchy subsets and attention-guided hierarchy aggregation (A-HA) module to highlight the key hierarchy edge sets with representative spatial average pooling and hierarchical edge convolution.

The DG-STGCN model (Dynamic Group Spatio-Temporal GCN) is proposed by Duan et al. [44]. DGSTGCN has the following advantages. The spatial modeling is built on learning the learnable coefficient matrices. The dynamic spatial-temporal modeling of the skeleton motion diversified groups of graph convolutions and temporal convolutions is designed dynamically group-wise.

The STGAT is proposed by Hu et al. [77] to capture short-term dependencies of spatial-temporal modeling. STGAT uses the three simple modules to reduce local spatial-temporal feature redundancy and further release the potential. STGAT builds local spatial-temporal graphs by connecting nodes in local spatial-temporal neighborhoods and dynamically constructing their relationships.

Duan et al. [78] proposed an open-source toolbox for skeleton-based action recognition based on PyTorch called PYSKL. PYSKL implements six different algorithms under a unified framework with both the latest and original good practices to ease the comparison of efficacy and efficiency. The PYSKL framework is built on top of ST-GCN, and PYSKL is the version of ST-GCN++.

The InfoGCN framework is proposed by Chi et al. [79] and presented in Figure 11. InfoGCN is a learning framework that combines a novel learning objective and an encoding method. The authors used the attention-based graph convolution that captures the context-dependent intrinsic topology of human action to learn the discriminative information for classifying action. A multi-modal representation of the skeleton using the relative position of joints also provides complementary spatial information for joints.

The TCA-GCN (Temporal-Channel Aggregation Graph Convolutional Networks) method is proposed by Wang et al. [80]. The TCA-GCN is used to learn spatial and temporal topologies dynamically and efficiently aggregate topological features in different temporal and channel dimensions for HAR. The TCA-GCN process of learning features is divided into two types: the TA module to learn temporal dimensional features and the channel aggregation module to efficiently combine spatial dynamic channel-wise topological features with temporal dynamic topological features.

#### 3.1.4. Hybrid-DNN

Hybrid-DNN approaches use deep learning networks together to extract features and train recognition models. Here we examine a series of studies from 2019 to 2023 for skeletal data-based activity recognition. Si et al. [81] proposed the AGC-LSTM (Attention Enhanced Graph Convolutional LSTM Network). The AGC-LSTM is capable of combining discriminative features in spatial configuration, temporal dynamics, and exploring the co-occurrence relationship between spatial and temporal domains. The AGC-LSTM also uses the AGC-LSTM layer to learn high-level semantic representation and significantly reduce the computation cost. The end-to-end trainable framework is proposed [82] with a combination of a Bayesian neural network (BNN) model where BNN is again combined from the graph convolution and LSTM. The graph convolution is used to capture the spatial dependency among different body joints and LSTM is used to capture the temporal dependency of pose change over time. A new SSNet (Scale Selection Network) is proposed [83] for online action prediction. SSNet learns the proper temporal window scale at each step to cover the performed part of the current action instance. The network predicts the ongoing action at each frame. Shi et al. [84] proposed a DSTA-Net (decoupled spatial-temporal attention networks). It is built with pure attention modules without manual designs of traversal rules or graph topologies. The spatial-temporal attention decoupling, decoupled position encoding, and spatial global regularization are used to build attention networks. The DSTA-Net model splits the skeleton’s data into four streams: spatial-temporal stream, spatial stream, slow-temporal stream, and fast-temporal stream, each focusing on a specific aspect of the skeleton sequence. These data streams are then combined to obtain a feature vector that best describes the skeleton data in space and time, as presented in Figure 12.

The end-to-end SGN network (Semantics-Guided Neural Network) is proposed [85] based on the combination of GCN and CNN models. It consists of a joint-level module and a frame-level module. SGN learns the dynamics representation of a joint by fusing the position and velocity information of a joint. To model the dependencies of joints in the joint-level module and the dependencies of frames, SGN used the three GCN layers and two CNN layers, respectively. Plizzaria et al. [86] proposed a novel two-stream Transformer-based model that is used on both the spatial and the temporal dimensions. The first stream is the Spatial Self-Attention (SSA) module to dynamically build links between skeleton joints, representing the relationships between human body parts, conditionally on the action and independently from the natural human body structure. The second stream is a Temporal Self-Attention (TSA) module to study the dynamics of a joint over time. Xiang et al. [87] employed a large-scale language model as the knowledge engine to provide text descriptions for body parts’ movements for actions. The authors proposed a multi-modal training scheme by utilizing the text encoder to generate feature vectors for different body parts and supervise the skeleton encoder for action representation learning. This means that the skeleton is divided into parts, and every human action is a working combination of the parts. Each part is coded with a descriptive text.

Trived et al. [88] proposed PSUMNet (Part Stream Unified Modality Network) for HAR based on human skeleton data. It introduces the combined modality part-based streaming approach compared to the conventional modality-wise streaming approaches. PSUMNet performs across skeleton action recognition datasets compared to state-of-the-art methods, yet it reduces the number of parameters by around 100–400%.

Zhou et al. [89] built a hybrid model named Hyperformer. This model used a solution to incorporate bone connectivity into Transformer via a graph distance embedding. Unlike GCN, which only uses the skeleton structure for initialization, Hyperformer retains the skeleton structure during training. Hyperformer also implements a self-attention (SA) mechanism on hypergraph, termed Hypergraph Self-Attention (HyperSA), to incorporate intrinsic higher-order relations into the model.

The action capsule network (CapsNet) for skeleton-based action recognition is proposed by Bavil et al. [90]. The temporal features associated with each joint are hierarchically encrypted based on ResTCN (Residual Temporal Convolution Neural Network) and CapsNet to focus on a set of critical joints dynamically. CapsNet learns to dynamically attend to features of pivotal joints and for each action of the human skeleton.

### 3.2. Datasets

To evaluate deep learning models for HAR based on 3D human skeleton data, usually, some benchmark datasets have to be used to evaluate the performance. Here we introduce some databases containing 3D human skeleton data.

**UTKinect-Action3D Dataset** [91] includes three types of data: color image, depth image, and 3D human skeleton. It is captured from a single MS Kinect with Kinect for Windows SDK Beta Version and 10 action types of human: walk, sit down, stand up, pick up, carry, throw, push, pull, wave hands, and clap hands. The skeleton data of each frame includes 20 joints, each joint has coordinates (x,y,z), and the total is 199 motion sequences.

**SBU-Kinect dataset** [92] is captured from the Microsoft Kinect sensor. It includes in total 282 video sequences belonging to eight classes of type actions: “approaching”, “departing”, “pushing”, “kicking”, “punching”, “exchanging objects”, “hugging”, and “hand shaking”. Each skeleton frame annotated 15 joints for each person by OpenNI with NITE middleware provided by PrimeSense, and each frame has 2 persons.

**Florence 3D Actions dataset** [93] is captured from MS Kinect sensors at the University of Florence in 2012. This dataset includes nine class activities: wave, drink from a bottle, answer the phone, clap, clinch, sit down, stand up, read, watch, and bow. The actions were performed 2/3 times by 10 subjects, resulting in 215 action sequences. Each skeleton frame includes 15 body joints (with *x*, *y*, and *z* coordinates) captured with MS Kinect.

**J-HMDB dataset** [94] is a subset of HMDB [95] with 21 action classes: brush hair, catch, clap, climb stairs, golf, jump, kickball, pick, pour, pull-up, push, run, shoot ball, shoot a bow, shoot a gun, sit, stand, swing baseball, throw, walk, wave. Each skeleton frame includes a total of 15 joints, of which there are 13 joints (left shoulder, right shoulder, left elbow, right elbow, left wrist, right wrist, left hip, right hip, left knee, right knee, left ankle, right ankle, neck) and two landmarks (face and belly). J-HMDB contains 928 samples and uses 3 train/test splits in the ratio of 7:3 (70% training and 30% testing).

**Northwestern UCLA Multiview Action 3D (N-UCLA)** [96] is captured by the MS Kinect version 1 sensor from various viewpoints. The training data are captured from view 1 and view 2, and the testing data are captured from view 3. This dataset includes 10 action categories: pick up with one hand, pick up with two hands, drop trash, walk around, sit down, stand up, donning, doffing, throwing, and carrying. Each action is performed by 10 subjects.

**SYSU 3D Human-Object Interaction Dataset** [97] includes 480 skeleton sequences with 12 action classes performed by 40 different subjects. The number of joints in each human skeleton is 20 joints. In each action, each subject can interact with one of six objects: phone, chair, bag, wallet, mop, and besom. For training and evaluation, the authors used two data split protocols as follows. The first protocol is for half of the samples for training and the other half for testing. The second protocol is to use half of the subjects for training and the other half for testing.

**NTU RGB+D dataset** [11] has been captured by three MS Kinect V2 sensors. 3D skeletal data contains the 3D locations/joints (with x,y,z coordinates) of 25 major body joints at each frame. It contains 56,880 skeleton videos of 60 action classes. This dataset is split in two ways: cross-subject and cross-view. The cross-subject includes 40,320 videos from 20 subjects for training and the rest for testing. The cross-view include 37,920 videos captured from camera 2 and 3 for training and those from camera 1 for testing.

**Kinetics-Skeleton dataset** [98] is named from DeepMind Kinetics human action video dataset. It includes 400 human action classes captured from nearly 300 videos. Each video is about 10 s, and the 3D human skeleton is annotated from the Open-Pose toolbox [99]. Each human skeleton includes 18 joints. The training and test sets of this dataset consist of 240, 436, and 19,794 samples, respectively.

**NTU RGB+D 120 dataset** [12] is extended from **NTU RGB+D dataset** [11]. Most descriptions of this dataset are the same as the **NTU RGB+D dataset** [11], only the number of action classes is 120, and the number of samples is expanded to 114,480. This dataset is split into two parts: an auxiliary set and a one-shot evaluation set. The auxiliary set contains 100 classes, and all samples of these classes can be used for training. The evaluation set consists of 20 novel classes, and one sample from each novel class is picked as the exemplar, while all the remaining samples of these classes are used to test the recognition performance. Two evaluation protocols for NTU RGB+D 120 dataset are set similarly to those in the NTU RGB+D dataset, where Cross-Subject in these two datasets has the same name, and Cross-View in the NTU-RGB+D is renamed Cross-Setting.

**KLHA3D-102 dataset** [100] is captured and combined from eight cameras of the MOCAP system. The 3D human skeleton of each frame includes 39 joints, as illustrated in Figure 13. In KLHA3D-102 consists of 102 classes, with each action class having five subjects (Sub), so the total video frame is 510 with 299,468 frames.

**KLYOGA3D dataset** [14] that the structure of the joints of the 3D human skeleton is similar to the **KLHA3D-102 dataset** [100]. The only difference, it has 39 action classes of yoga skeletal, so the total number of video frames is 39 with 173,060 frames.

### 3.3. Evaluation Matrices

To evaluate and compare the performance of HAR models based on a 3D human skeleton, the measurements are essential and must be unified. Usually, in machine learning, model evaluation is often based on the accuracy metric as follows:Accuracy (*Acc*):
(1)$$Acc=\frac{TP+TN}{TP+TN+FP+FN}$$
where TP (True Positive) is the number of predictions when the label is positive and the prediction is true, TN (True Negative) is the number of predictions when the label is negative and the prediction is true, FP (False Positive) is the number of predictions when the label is positive, but the prediction is false, FN (False Negative) is the number of predictions when the label is negative, but the prediction is false.

### 3.4. Literature Results

The results of HAR based on the 3D human skeleton data of the NTU RGB+D [11] dataset are presented in Table 1. We presented the results of 66 valuable studies over time from 2019 to March 2023. The results are based on the *Acc* measure and two protocols: Cross-Subject and Cross-View [11].

The results of HAR based on the 3D human skeleton data of the NTU RGB+D 120 [12] dataset are presented in Table 2. We present the results of 37 valuable studies over time from 2019 to March 2023. The results are based on the *Acc* measure and two protocols: Cross-Subject and Cross-Setting [11].

In Table 3, we have shown the number of FLOPS (floating point operation per second) for training and testing of some DNNs on the NTU RGB + D [11] and NTU RGB + D 120 [12] datasets. With the larger number of FLOPS, the processing speed of the DNNs is slower.

The results of HAR based on the 3D human skeleton data of the Kinetics-Skeleton dataset [98] are presented in Table 4. We present the results of 17 studies over the period from 2019 to March 2023. The results are based on the *Acc* measure with the training and test dataset presented in [98].

Table 5 presents the results of HAR based on the 3D human skeleton data of the N-UCLA dataset [96]. We presented the results of 17 studies over the period from 2019 to March 2023. The results are based on the *Acc* measure with the training and test dataset presented.

Table 6 presents the results of HAR based on the 3D human skeleton data of the J-HMDB dataset [94]. Studies on HAR based on the 3D human skeleton on the J-HMDB dataset are only available from the year 2019.

Table 7 presents the results of HAR based on the 3D human skeleton data of the SYSU 3D dataset [97]. Studies on HAR based on a 3D human skeleton on the SYSU 3D dataset are only available from the years 2019 and 2020.

Table 8 presents the results of HAR based on the 3D human skeleton data of the UTKinect-Action3D dataset [91]. In the UTKinect-Action3D dataset [91], the authors used the “Test Two” protocol in [123] for training and testing (2/3 of the samples were for training, the rest for testing). Studies on HAR based on a 3D human skeleton on the UTKinect-Action3D dataset are only available in the years 2019 and 2020.

Table 9 presents the results of HAR based on the 3D human skeleton data of the Florence 3D Actions dataset [93]. In the Florence 3D Actions dataset, [93] have the same protocol for training and evaluation as the UTKinect-Action3D dataset [91].

Table 10 presents the results of HAR based on the 3D human skeleton data of the SBU dataset [92]. The training and testing data have been described in [92].

### 3.5. Challenges and Discussion

In Table 1, Table 2, and Table 4, Table 5, Table 6, Table 7, Table 8, Table 9 and Table 10, we presented the results of HAR based on a 3D human skeleton. The datasets of 3D human skeletons are presented in Section 3.2; in each scene/frame in 3D space, only one 3D human skeleton is considered. The activities identified are simple and common everyday activities. The datasets presented are of no interest and contain data on the context of human activity. Therefore, the building recognition models are only used for testing but have not been able to apply it in practice to build real applications.

These tables also show that the GCN/GNN-based approach is the most interesting in the research because the structure of the human skeleton is represented as graphs, and the temporal-spatial functions are important information to represent and extract feature vectors. Table 1 and Table 2 presented the results on the NTU RGB + D [11] and NTU RGB + D 120 [12], respectively. Although it is the same evaluation method and measure, the results on the NTU RGB + D dataset are much higher than on the NTU RGB + D 120 dataset. It can be seen that the number of action classes in the data greatly affects the results. It increases the complexity of the activity recognition problem. In the Kinetics-Skeleton dataset [98], the number of human action classes is 400 classes, so the results in Table 4 are very low. The highest result is 52.3% when there is a combination of the 3D skeleton and RGB images. The results only based on 3D human skeletons are only 30–40%.

## 4. Comparative Study of HAR

### 4.1. Experiment

In this paper, we perform a comparative study on HAR based on the 3D human skeleton alone. This study was conducted on the two databases presented above, namely the KLHA3D-102 dataset [100] and the KLYOGA3D dataset [14]. We use DDNet [120] and PA-ResGCN [59] to experiment on two datasets.

We divide the KLHA3D-102 dataset [100] into five configurations for training and testing: Configuration 1 (KLHA3D-102_Conf. 1) has Sub #2, Sub #3, Sub #4, and Sub #5 of each action used for training, and Sub #1 for testing; Configuration 2 (KLHA3D-102_Conf. 2) has Sub #1, Sub #3, Sub #4, and Sub #5 of each action used for training and Sub #2 for testing; Configuration 3 ((KLHA3D-102_Conf. 3) has Sub #1, Sub #2, Sub #4, and Sub #5 of each action used for training and Sub #3 for testing; Configuration 4 (KLHA3D-102_Conf. 4) has 15% of the first frames of each sub of each action used for testing and the remaining 85% of frames in each sub of each action for training; Configuration 5 (KLHA3D-102_Conf. 5) has 85% of the first frames of each sub of each action used for training and 15% of the remaining frames in each sub of each action for testing.

We also divide the KLYOGA3D dataset [14] into two configurations for training and testing: Configuration 1 (KLYOGA3D_Conf. 1) has 15% of the first frames of each sub of each action used for testing, and the remaining 85% of frames in each sub of each action for training; Configuration 2 (KLYOGA3D_Conf. 2) has 85% of the first frames of each sub of each action used for training and 15% of the remaining frames in each sub of each action for testing.

In this paper, we used a server with an NVIDIA GeForce RTX 2080 Ti 12 GB GPU for fine-tuning, training, and testing. The programs were written in the Python language (≥3.7 version) with the support of CUDA 11.2/cuDNN 8.1.0 libraries. In addition, there are a number of other libraries such as OpenCV, Numpy, Scipy, Pillow, Cython, Matplotlib, Scikit-image, Tensorflow ≥ 1.3.0, etc.

In this paper, we pre-trained the models of DDNet [120] and PA-ResGCN [59] with 500 epochs. The HAR results are compared with CNN-LSTM [128], SgCNN [14], and CCNN [129].

### 4.2. Results and Discussion

The results of HAR based on the skeleton of the KLHA3D-102, KLYOGA3D datasets are shown in Table 11. The results show that GCN-based DNNs and Hybrid DNNs are very low (DDnet when evaluated on KLHA3D-102_Conf. 5 = 1.96%, PA-ResGCN when evaluated on KLHA3D-102_Conf. 5 = 8.56%). Meanwhile, the results based on CNNs have very high results compared to GCN/GNN and Hybrid-DNN networks.

This can be explained by several reasons. The model is trained to recognize using only the data of the 3D human skeleton and has not been combined with other data, such as data about the activity context. Human skeleton data are collected from many cameras in different viewing directions without being normalized. The number of joints in the skeleton data of these two datasets are 39 joints, which is a large number of joints in 3D space. They make the feature vector size large, and the number of action classes in the KLHA3D-102 dataset is 102 classes; there are many similar actions, such as “drinking tea”, “drinking water”, and “eating”. As illustrated by the human skeleton data in Figure 1 of the KLHA3D-102, KLYOGA3D datasets. The actions “drinking tea”, “drinking water”, and “eating” differ only in the coordinates of the 3 joints (“14”, “15”, “16”) or (“21”, “22”, “23”). Therefore, the size of the feature to distinguish between these three types of actions is too small ($\frac{3}{39}=\frac{1}{13}$) compared to the size of the entire feature extracted from the 39 joints of the 3D human skeleton. These actions are all sitting and holding/grasping a bowl or cup, the action is only slightly different in the hand activity. At least the computational complexity will be reduced by $\frac{1}{3}$ times when switching from computing features in 3D space to 2D space (image space). This is also the reason Kumar et al. [14] chooses the approach of projecting the representation of the feature vectors of the KLHA3D-102 and KLYOGA3D datasets from the 3D space to the image space. All these make the features extracted from the 3D skeleton based on the skeleton graph and the space-time function have low discrimination. That makes the result of active recognition based on GCN and Hybrid-DNN lower than CNNs. CNNs often project representations (joints, coordinates, temporal-spatial) on the skeleton to the image space. This results in better discrimination between activities, as shown by the difference between the feature vectors of 10 joints in Figure 5 of Kumar et al.’s research [14]. In this figure, the top is a representation of JADM (Joint Angular Displacement Maps) [130], the middle is a representation of JDM (Joint Distance Maps) [131], and the bottom is a representation of QJVM (Quad Joint Volume Maps) [14].

In Table 12, the processing time to recognize the human activity on the KLHA3D-102 [100] dataset. The computation time of DDnet [120] is 100 times faster than PA-ResGCN [59].

Figure 14 shows the results of the training set and testing set on the KLHA3D-102 and KLYOGA3D datasets using DDNet [120]. On the KLHA3D-102 dataset, the result of DDnet on the training set is more than 80%, and the result on the testing set is just over 50%. On the KLYOGA3D dataset, the result on the training set is only more than 50%, and on the testing set, it is only more than 20%. This shows that the efficiency of learning features extracted directly on 3D human pose on the KLYOGA3D dataset is very low. When training DDnet on the KLHA3D-102 dataset, the result on the training dataset is more than 90%, but the result on the test dataset is about 50%. This result occurs because the trained model of DDnet is overfitting. Figure 15 illustrates the 3D human skeleton of the action of “drinking tea” and “drinking water”. The skeleton data of the two actions are almost the same; only the activity is different in the skeleton of the head. This is a huge challenge to build a discriminative model of human actions in the KLHA3D-102 dataset.

## 5. Conclusions and Future Works

In this paper, we have carried out a full survey of the methods of using deep learning to recognize human activities based on 3D human skeleton input data. Our survey produced about 250 results on about more than 70 different studies on HAR based on deep learning under four types of networks: RNN-based, CNN-based, GCN/GNN-based, and Hybrid-DNN-based. The results of HAR are shown in terms of methods and processing time. We also discuss the challenges of HAR in terms of data dimensions and the insufficient information to distinguish actions with a limited number of reference points. At the same time, we have carried out comparative, analytical, and discussion studies based on fine-tuning two methods of DNNs (DDNet, PA-ResGCN) for HAR on the KLHA3D-102 and KLYOGA3D datasets. Although the training set rate is up to 85% and the test set rate is 15%, the recognition results are still very low (the results on KLHA3D-102_Conf. 5 is 1.96% of DDnet and 8.56% of PA-ResGCN). It also shows that choosing a method for the HAR problem is very important; for datasets with a large number of joints in the 3D human skeleton, the method based on projecting a 3D human skeleton to the image space and extraction features on the image space should be chosen.

Shortly, we will combine many types of features extracted from the 3D human skeleton into a deep learning model or construct new 2D feature sets to improve higher HAR results. We will propose a unified model from end-to-end for detecting, segmenting, estimating 3D human pose, and recognizing human activities for training and learning exercises in the gym or yoga for training and protecting health. As illustrated in Figure 16 is an application that detects, segments, estimates 3D human pose, recognizes activity, and calculates the total motion of joints. From there, it is possible to calculate the total energy consumed for exercise. From there, it is possible to make a training plan for students to practice to protect their health, to avoid exercising too much or doing too little. This is a very practical application in martial arts teaching, sports analysis, training, and health protection.

## Figures and Tables

**Figure 1 sensors-23-05121-f001:**
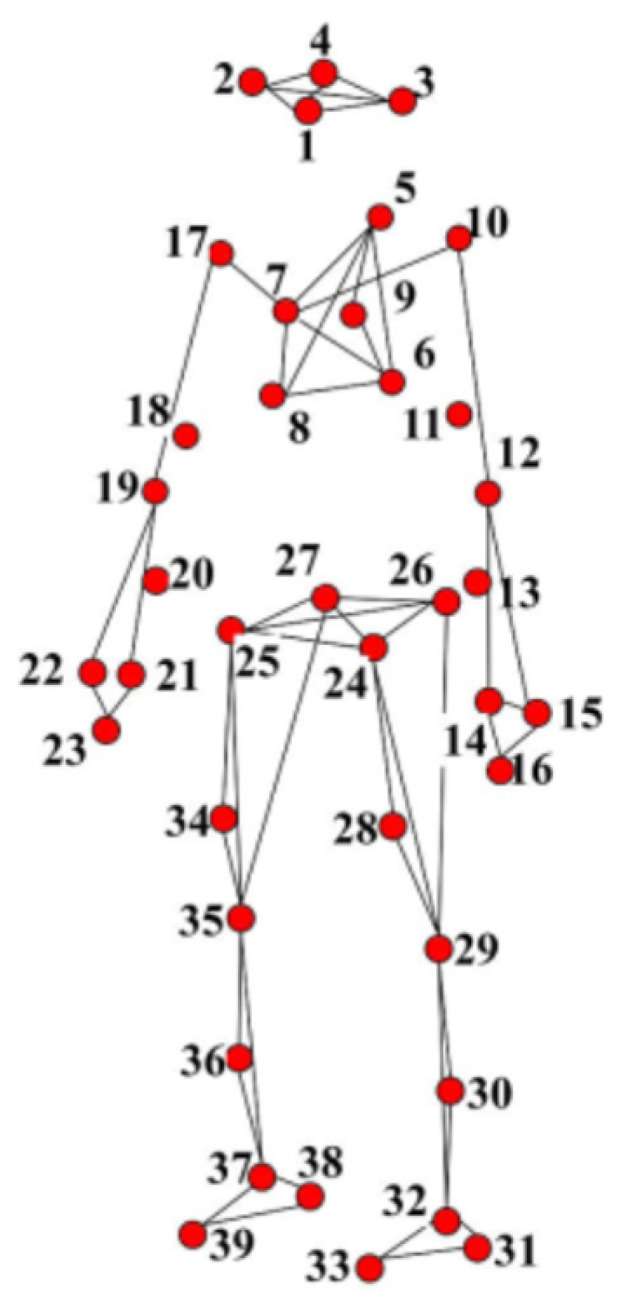
Illustrating 3D human skeleton data of the KLHA3D102 [14] and KLYOGA3D [14] datasets.

**Figure 2 sensors-23-05121-f002:**
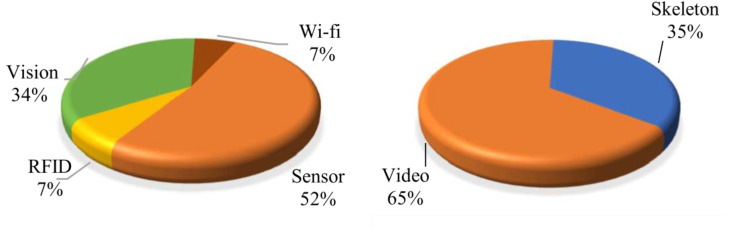
The left shows the proportion of HAR research based on vision information, and the right shows the percentage of input information for HAR from video or skeleton.

**Figure 3 sensors-23-05121-f003:**
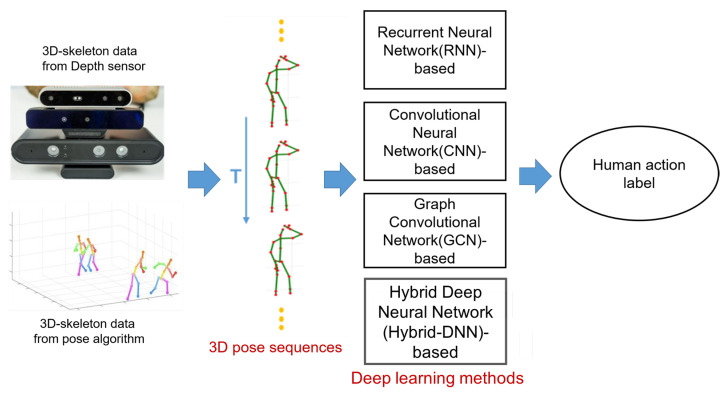
Illustrating four deep learning-based methods of HAR based on 3D human skeleton data.

**Figure 4 sensors-23-05121-f004:**
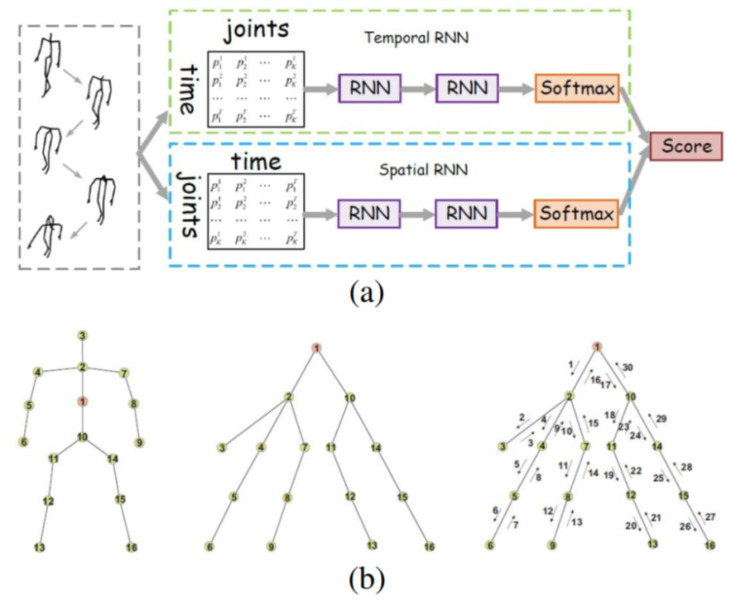
Illustrating the RNN-based approach for HAR based on 3D human skeleton [6]. (**a**) is the skeleton representation as a function of the temporal-spatial, (**b**) is the skeleton representation according to the tree structure.

**Figure 5 sensors-23-05121-f005:**
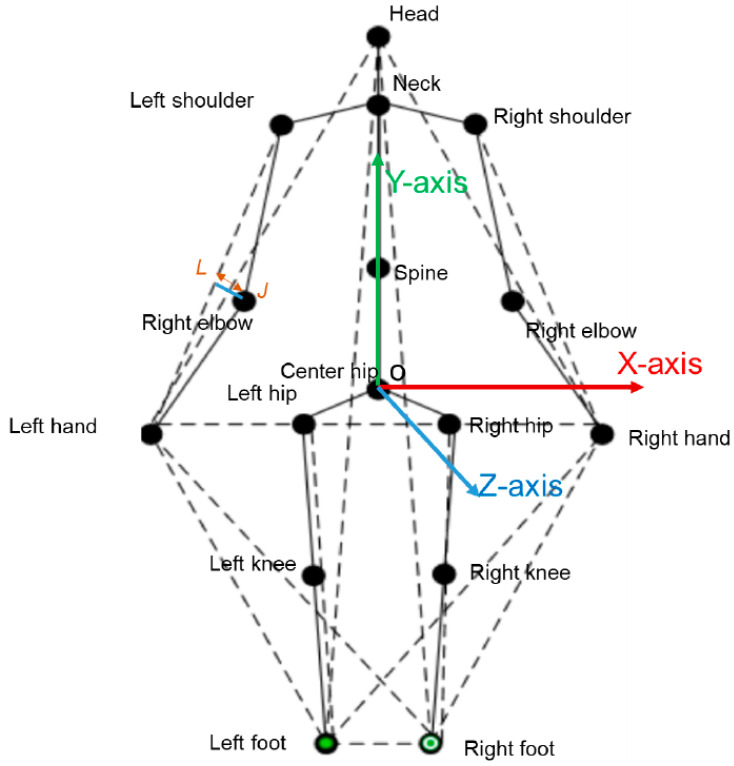
Illustrating a new coordinate system and representation of feature vectors based on joints of human body parts.

**Figure 6 sensors-23-05121-f006:**
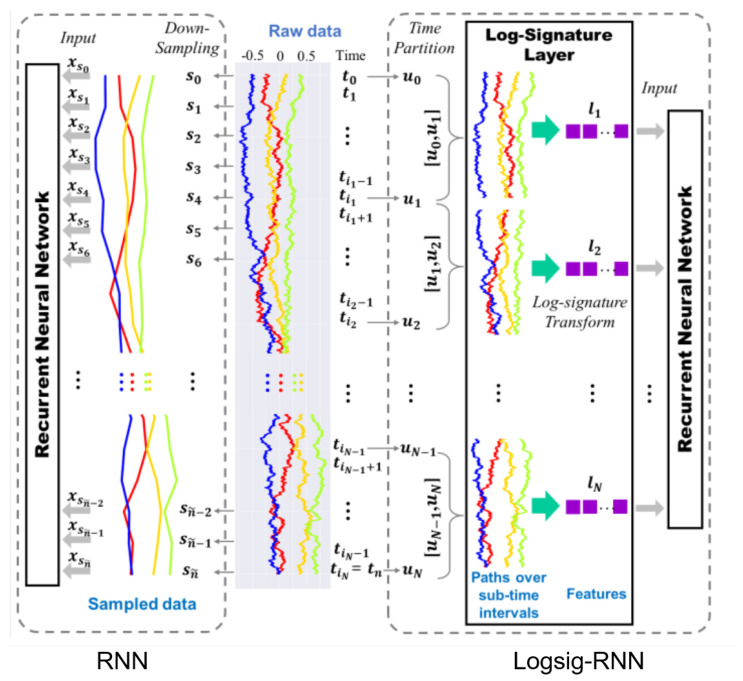
Illustrating the improved model of Logsig-RNN compared to RNN [40].

**Figure 7 sensors-23-05121-f007:**
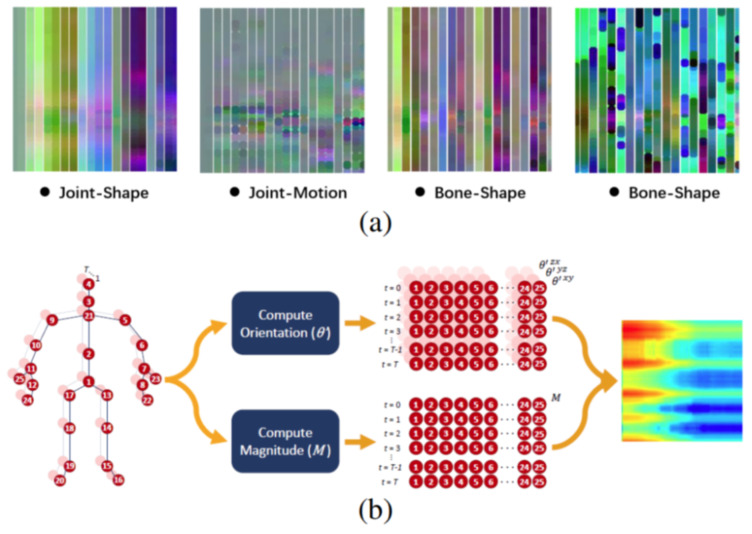
Illustrating CNN-based HAR [6]. (**a**) represents the feature types in the image space, and (**b**) represents the process of projecting the data of the 3D skeleton into the image space.

**Figure 8 sensors-23-05121-f008:**
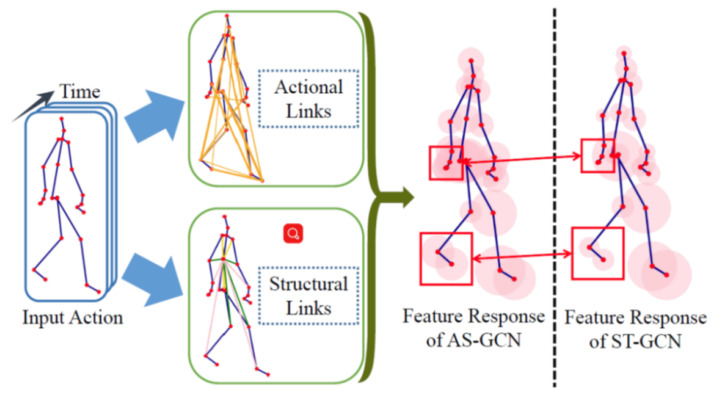
Illustrating feature extraction of GCN-based methods [6].

**Figure 9 sensors-23-05121-f009:**
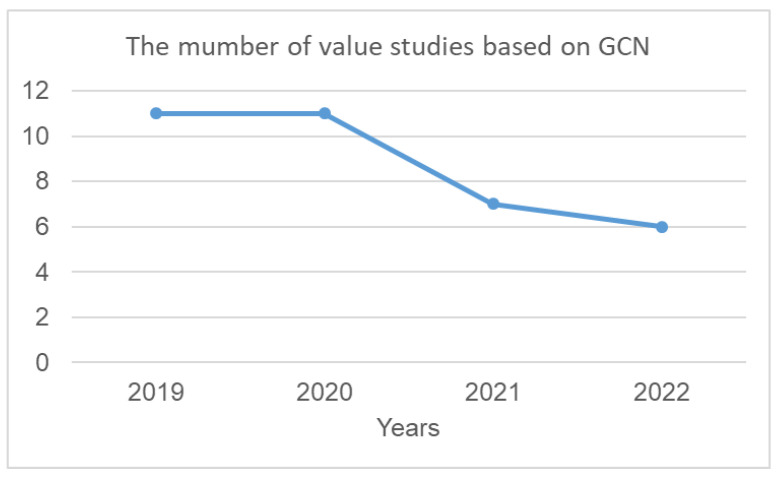
Statistics on the number of value studies based on GCN in the past four years.

**Figure 10 sensors-23-05121-f010:**
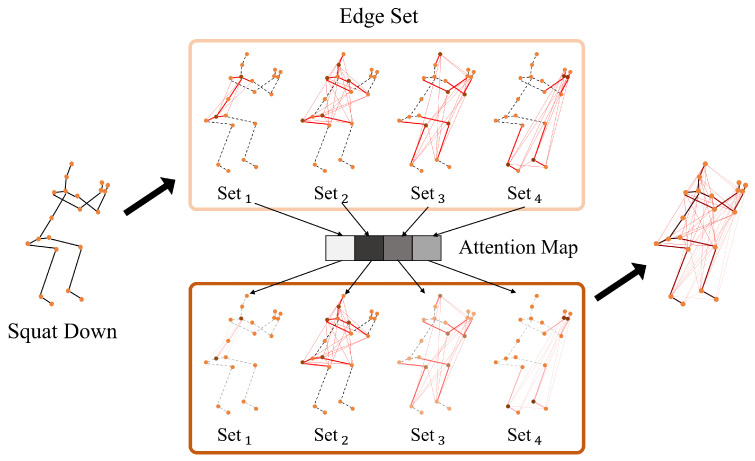
The illustration of the HD-GCN architecture [76].

**Figure 11 sensors-23-05121-f011:**
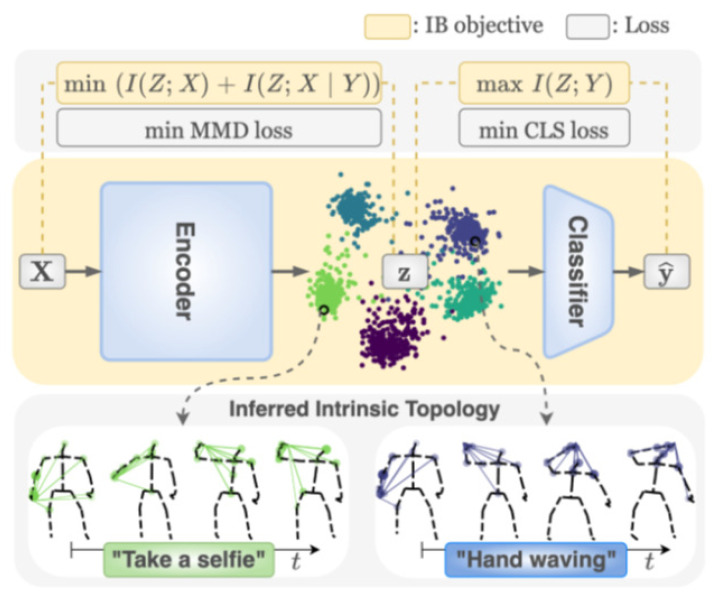
The InfoGCN framework [79].

**Figure 12 sensors-23-05121-f012:**
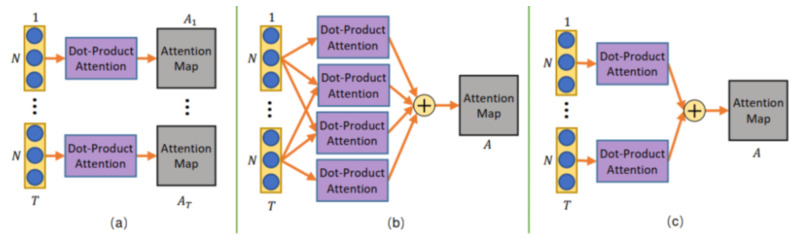
The DSTA-Net framework [84]. (**a**) is to compute the attention maps frame by frame, (**b**) is the calculation of the relations of two joints between all of the frames, (**c**) is a compromise.

**Figure 13 sensors-23-05121-f013:**
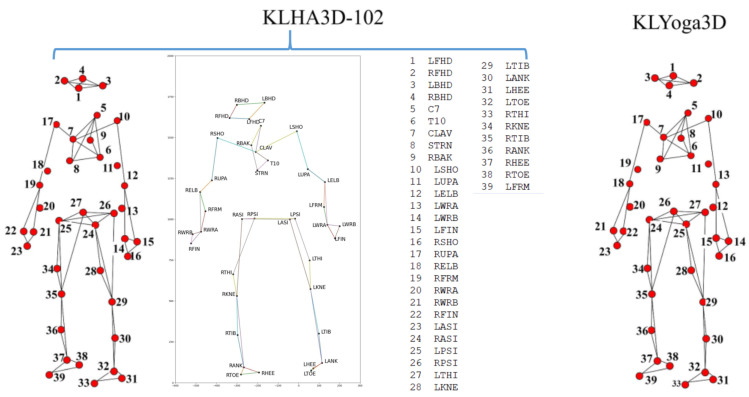
Illustrating the KLHA3D-102 and KLYOGA3D datasets.

**Figure 14 sensors-23-05121-f014:**
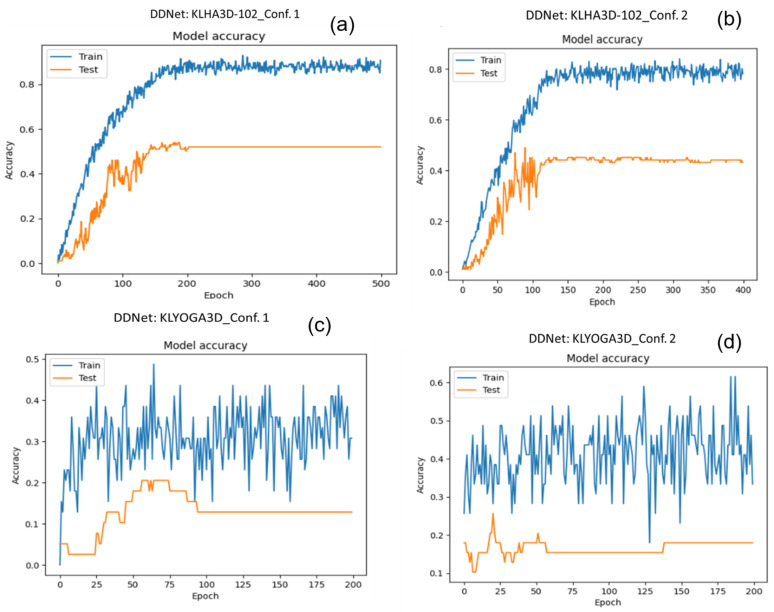
Illustrating the training and testing results on the KLHA3D-102 and KLYOGA3D datasets. (**a**) is the accuracy of the model on the train set and the test set of Conf.1 on the KLHA3D-102 dataset. (**b**) is the accuracy of the model on the train set and the test set of Conf.2 on the KLHA3D-102 dataset. (**c**) is the accuracy of the model on the train set and the test set of Conf.1 on the KLYOGA3D dataset. (**d**) is the accuracy of the model on the train set and the test set of Conf.2 on the KLYOGA3D dataset.

**Figure 15 sensors-23-05121-f015:**
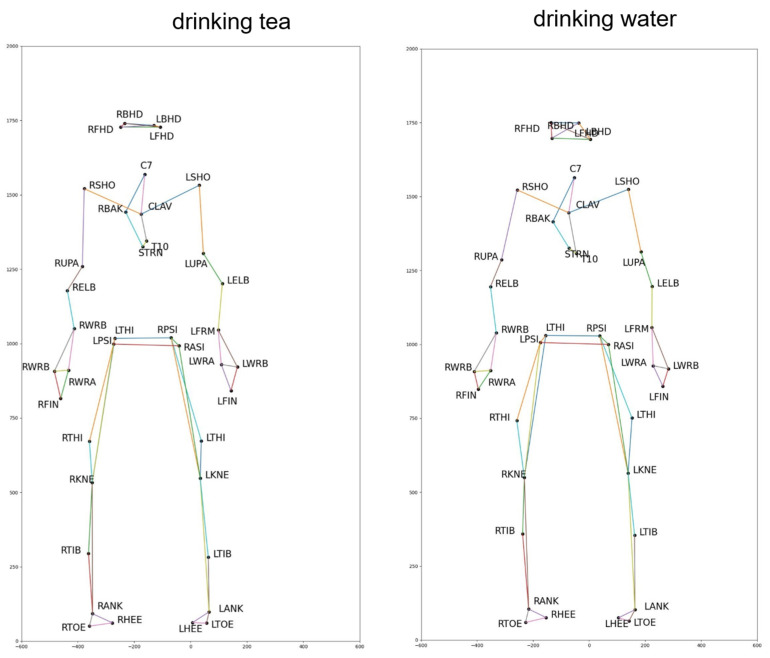
3D human skeleton illustration of “drinking tea” and “drinking wate” actions on the KLHA3D-102 dataset.

**Figure 16 sensors-23-05121-f016:**
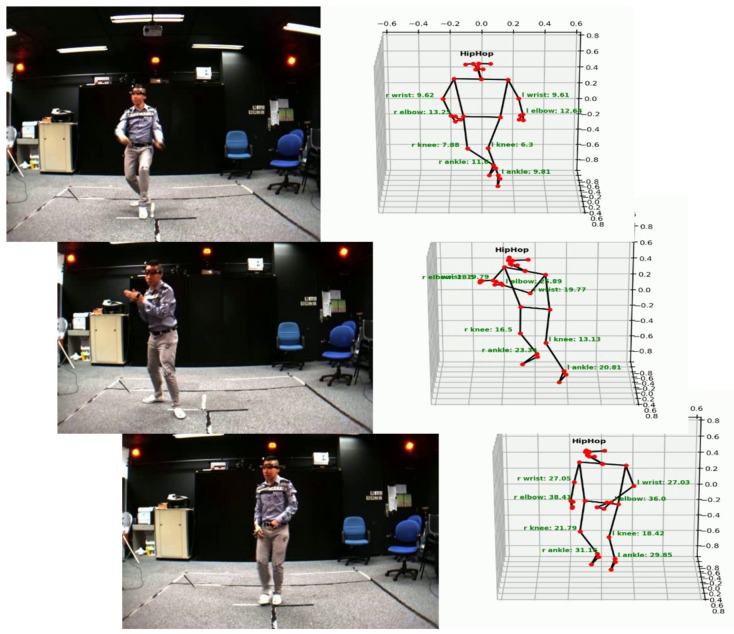
Illustrating application of 3D human pose estimation, activity recognition, and total distance traveled of joints on the human body.

**Table 1 sensors-23-05121-t001:** The results of HAR on the NTU RGB + D [11] dataset.

Authors	Years	Models	Cross-Subject *Acc* (%)	Cross-View *Acc* (%)	Type of DLMs
Wen et al. [101]	2019	Motif-STGCN	84.2	90.2	CNN
Song et al. [61]	2019	RA-GCN	85.9	93.5	GCN
Li et al. [39]	2019	DenseIndRNN	86.7	93.7	RNN
Li et al. [50]	2019	AS-GCN	86.8	94.2	GCN
Si et al. [81]	2019	AGC-LSTM (Joint & Part)	89.2	95	Hybrid-DNN
Lei et al. [102]	2019	2s-AGCN	88.5	95.1	GCN
Wu et al. [54]	2019	2s-SDGCN	89.6	95.7	GCN
Shi et al. [49]	2019	DGNN	89.9	96.1	GNN
Peng et al. [48]	2019	GCN-NAS	89.4	95.7	GCN
Gao et al. [52]	2019	BAGCN	90.3	96.3	GCN
Li et al. [53]	2019	Sym-GNN	90.1	96.4	GNN
Shi et al. [47]	2019	MS-AAGCN	90	96.2	GCN
Shi et al. [47]	2019	JB-AAGCN	89.4	96	GCN
Si et al. [81]	2019	AGC-LSTM	89.2	95	Hybrid-DNN
Liang et al. [103]	2019	3SCNN	88.6	93.7	CNN
Shi et al. [55]	2019	2s-NLGCN	88.5	95.1	GCN
Cho et al. [104]	2019	TS-SAN	87.2	92.7	Hybrid-DNN
Li et al. [105]	2019	RF-Action	86.8	91.6	Hybrid-DNN
Song et al. [61]	2019	3s RA-GCN	85.9	93.5	GCN
Song et al. [61]	2019	2s RA-GCN	85.8	93	GCN
Papadopoulos et al. [56]	2019	GVFE + AS-GCN with DH-TCN	85.3	92.8	GCN
Ding et al. [51]	2019	AR-GCN	85.1	93.2	GCN
Wang et al. [106]	2019	ST-GCN-jpd	83.36	88.84	GCN
Zhao et al. [82]	2019	Bayesian GC-LSTM	81.8	89	Hybrid-DNN
Zhang et al. [107]	2019	EleAtt-GRU	80.7	88.4	RNN
Caetano et al. [108]	2019	Skelemotion + Yang et al.	76.5	84.7	CNN
Caetano et al. [109]	2019	TSRJI	73.3	80.3	CNN
Zhang et al. [85]	2020	SGN	89	94.5	Hybrid-DNN
Wang et al. [110]	2020	MV-IGNET	89.2	96.3	Hybrid-DNN
Cheng et al. [60]	2020	4s Shift-GCN	90.7	96.5	GCN
Cheng et al. [111]	2020	DecoupleGCN-DropGraph	90.8	96.6	GCN
Song et al. [59]	2020	PA-ResGCN-B19	90.9	96	GCN
Liu et al. [62]	2020	MS-G3D	91.5	96.2	GCN
Koniusz et al. [45]	2020	SCK⊕	91.56	94.75	CNN
Shi et al. [84]	2020	DSTA-Net	91.5	96.4	Hybrid-DNN
Ye et al. [63]	2020	Dynamic GCN	91.5	96	GCN
Obinata et al. [64]	2020	MS-AAGCN + TEM	91	96.5	GCN
Yang et al. [70]	2020	CGCN	90.3	96.4	GCN
Yang et al. [75]	2020	FGCN-spatial + FGCN-motion	90.2	96.3	GCN
Plizzaria et al. [86]	2020	ST-TR-agcn	89.9	96.1	Hybrid-DNN
Peng et al. [48]	2020	Mix-Dimension	89.7	96	GCN
Yang et al. [65]	2020	PGCN-TCA	88	93.6	GCN
Song et al. [61]	2020	3s RA-GCN	87.3	93.6	GCN
Ding et al. [66]	2020	Sem-GCN	86.2	94.2	GCN
Yu et al. [67]	2020	PeGCN	85.6	93.4	GCN
Li et al. [68]	2020	PR-GCN	85.2	91.7	GCN
Fan et al. [112]	2020	RGB+Skeleton	84.23	89.27	GCN
Song et al. [74]	2021	EfficientGCN-B4	91.7	95.7	GCN
Chen et al. [71]	2021	CTR-GCN	92.4	96.8	GCN
Chi et al. [79]	2021	InfoGCN	93	97.1	GCN
Chen et al. [71]	2021	CTR-GCN	92.4	96.8	GCN
Chen et al. [69]	2021	DualHead-Net	92	96.6	GCN
Qin et al. [72]	2021	AngNet-JA + BA + JBA + VJBA	91.7	96.4	GCN/GNN
Zeng et al. [73]	2021	Skeletal GNN	91.6	96.7	GNN
Song et al. [74]	2021	EfficientGCN-B2	90.9	95.5	GCN
Song et al. [74]	2021	EfficientGCN-B0	89.9	94.7	GCN
Duan et al. [44]	2022	PoseC3D	94.1	97.1	CNN
Trivedi et al. [88]	2022	PSUMNet	92.9	96.7	Hybrid-DNN
Lee et al. [76]	2022	HD-GCN	93.4	97.2	GCN
Duan et al. [44]	2022	DG-STGCN	93.2	97.5	GCN
Xiang et al. [87]	2022	LST	92.9	97	Hybrid-DNN
Hu et al. [77]	2022	STGAT	92.8	97.3	GCN
Wang et al. [80]	2022	TCA-GCN	92.8	97	GCN
Duan et al. [78]	2022	ST-GCN++ [PYSKL, 3D Skeleton]	92.6	97.4	GCN
Duan et al. [78]	2022	ST-GCN [PYSKL, 2D Skeleton]	92.4	98.3	GCN
Zhou et al. [89]	2023	Hyperformer	92.9	96.5	Hybrid-DNN
Bavil et al. [90]	2023	Action Capsules	90	96.3	Hybrid-DNN

**Table 2 sensors-23-05121-t002:** The results of HAR on the NTU RGB + D 120 [12] dataset.

Authors	Years	Models	Cross-Subject *Acc* (%)	Cross-Setting *Acc* (%)	Type of DLMs
Caetano et al. [108]	2019	SkeleMotion (Magnitude-Orientation)	62.9	63	CNN
Caetano et al. [108]	2019	SkeleMotion + Yang et al	67.7	66.9	CNN
Caetano et al. [109]	2019	TSRJI	67.9	59.7	CNN
Song et al. [61]	2019	3s RA-GCN	81.10	82.70	GCN
Papadopoulos et al. [56]	2019	GVFE + AS-GCN with DH-TCN	78.30	79.80	GCN
Liao et al. [40]	2019	Logsig-RNN	68.30	67.20	RNN
Liu et al. [83]	2019	FSNet	59.90	62.40	Hybrid-DNN
Zhang et al. [85]	2020	SGN	79.2	81.5	Hybrid-DNN
Cheng et al. [60]	2020	4s Shift-GCN	85.9	87.6	GCN
Cheng et al. [111]	2020	DecoupleGCN-DropGraph	86.5	88.1	GCN
Liu et al. [62]	2020	MS-G3D	86.9	88.4	GCN
Song et al. [59]	2020	PA-ResGCN-B19	87.3		
Shi et al. [84]	2020	DSTA-Net	86.6	89.0	Hybrid-DNN
Yang et al. [75]	2020	FGCN-spatial + FGCN-motion	85.4	87.4	GCN
Plizzaria et al. [86]	2020	ST-TR-agcn	82.70	84.70	Hybrid-DNN
Peng et al. [48]	2020	Mix-Dimension	80.50	83.20	GCN
Memme et al. [113]	2020	Gimme Signals	70.80	71.60	CNN
Song et al. [74]	2021	EfficientGCN-B4	88.3	89.1	GCN
Chen et al. [71]	2021	CTR-GCN	88.9	90.6	GCN
Chen et al. [71]	2021	InfoGCN	89.8	91.2	GCN
Chen et al. [69]	2021	DualHead-Net	88.2	89.3	GCN
Qin et al. [72]	2021	AngNet-JA + BA + JBA + VJBA	88.2	89.2	GCN/GNN
Song et al. [74]	2021	EfficientGCN-B2	87.90	88.00	GCN
Zeng et al. [73]	2021	Skeletal GNN	87.5	89.2	GNN
Song et al. [74]	2021	EfficientGCN-B0	85.90	84.30	GCN
Duan et al. [44]	2022	PoseC3D	86.9	90.3	CNN
Trivedi et al. [88]	2022	PSUMNet	89.4	90.6	Hybrid-DNN
Lee et al. [76]	2022	HD-GCN	90.1	91.6	GCN
Xiang et al. [87]	2022	LST	89.9	91.1	Hybrid-DNN
Duan et al. [44]	2022	DG-STGCN	89.6	91.3	GCN
Wang et al. [80]	2022	TCA-GCN	89.4	90.8	GCN
Hu et al. [77]	2022	STGAT	88.7	90.4	GCN
Duan et al. [78]	2022	ST-GCN++ [PYSKL, 3D Skeleton]	88.6	90.8	GCN
Zhou et al. [89]	2023	Hyperformer	89.9	91.3	Hybrid-DNN

**Table 3 sensors-23-05121-t003:** The number of FLOPS when training on the NTU RGB+D [11] and NTU RGB + D 120 [12] datasets.

Models	FLOPs	Type of DLMs
TaCNN+ [114]	1.0	GCN/GNN
ST-GCN [16]	16.3	GCN/GNN
RA-GCN [61]	32.8	GCN/GNN
2s-AGCN [102]	37.3	GCN/GNN
PA-ResGCN [59]	18.5	GCN/GNN
4s-ShiftGCN [60]	10.0	GCN/GNN
DC-GCN+ADG [111]	25.7	GCN/GNN
CTR-GCN [71]	7.6	GCN/GNN
DSTA-Net [84]	64.7	Hyprid-DNN
ST-TR [86]	259.4	Hyprid-DNN
PSUMNet [88]	2.7	Hyprid-DNN

**Table 4 sensors-23-05121-t004:** The results of HAR on the Kinetics-Skeleton [98] dataset.

Authors	Years	Models	Activity Recognition *Acc* (%)	Type of DLMs
Lei et al. [102]	2019	2s-AGCN	38.6	GCN
Shi et al. [47]	2019	MS-AAGCN	37.8	GCN
Shi et al. [47]	2019	JB-AAGCN	37.4	GCN
Peng et al. [48]	2019	GCN-NAS	37.1	GCN
Shi et al. [49]	2019	DGNN	36.9	GNN
Li et al. [50]	2019	AS-GCN	34.8	GCN
Li et al. [115]	2019	ST-GR	33.6	GCN
Ding et al. [51]	2019	AR-GCN	33.5	GCN
Liu et al. [62]	2020	MS-G3D	38	GCN
Ye et al. [63]	2020	Dynamic GCN	37.9	GCN
Yang et al. [70]	2020	CGCN	37.5	GCN
Plizzaria et al. [86]	2020	ST-TR-agcn	37.4	Hybrid-DNN
Yu et al. [67]	2020	PeGCN	34.8	GCN
Li et al. [68]	2020	PR-GCN	33.7	GCN
Chen et al. [69]	2021	DualHead-Net	38.4	GCN
Duan et al. [44]	2022	PoseC3D	49.1	CNN
Hachiuma et al. [116]	2023	Structured Keypoint Pooling	52.3	CNN

**Table 5 sensors-23-05121-t005:** The results of HAR on the N-UCLA [96] dataset.

Authors	Years	Models	Activity Recognition *Acc* (%)	Type of DLMs
Zhang et al. [107]	2019	EleAtt-GRU	90.7	RNN
Davoodikakhki et al. [117]	2020	Hierarchical Action Classification	93.99	CNN
Zhang et al. [85]	2020	SGN	92.5	Hybrid-DNN
Chi et al. [79]	2021	InfoGCN	97	GCN
Chen et al. [71]	2021	CTR-GCN	96.5	GCN
Xiang et al. [87]	2022	LST	97.2	Hybrid-DNN
Lee et al. [76]	2022	HD-GCN	97.2	GCN
Wang et al. [80]	2022	TCA-GCN	97	GCN
Bavil et al. [90]	2023	Action Capsules	97.3	Hybrid-DNN

**Table 6 sensors-23-05121-t006:** The results of HAR on the J-HMDB [94] dataset.

Authors	Years	Models	Activity Recognition Accuracy (%)	Type of DLMs
Yan et al. [118]	2019	PA3D+RPAN	86.1	CNN
Nally et al. [119]	2019	STAR-Net	64.3	CNN
Yang et al. [120]	2019	DD-Net	77.2	Hybrid-DNN
Ludl et al. [121]	2019	EHPI	65.5	CNN

**Table 7 sensors-23-05121-t007:** The results of HAR on the SYSU 3D [97] dataset.

Authors	Years	Models	Activity Recognition *Acc* (%)	Type of DLMs
Zhang et al. [107]	2019	EleAtt-GRU	85.7	RNN
Ke et al. [122]	2019	Local + LGN	83.14	Hybrid-DNN
Zhang et al. [85]	2020	SGN	86.9	Hybrid-DNN

**Table 8 sensors-23-05121-t008:** The results of HAR on the UTKinect-Action3D [91] dataset.

Authors	Years	Models	Activity Recognition *Acc* (%)	Type of DLMs
Kao et al. [57]	2019	GFT	96.0	GCN
Paoletti et al. [124]	2020	Temporal Subspace Clustering	99.5	Hybrid-DNN
Koniusz et al. [45]	2020	SCK ⊕ DCK	99.2	CNN

**Table 9 sensors-23-05121-t009:** The results of HAR on the Florence 3D Actions [93] dataset.

Authors	Years	Models	Activity Recognition *Acc* (%)	Type of DLMs
Koniusz et al. [45]	2020	SCK ⊕ + DCK⊕	97.45	CNN
Paoletti et al. [124]	2020	Temporal Spectral Clustering + Temporal Subspace Clustering	95.81	Hybrid-DNN
Koniusz et al. [45]	2020	SCK + DCK	95.23	CNN

**Table 10 sensors-23-05121-t010:** The results of HAR on the SBU [92] dataset.

Authors	Years	Models	Activity Recognition *Acc* (%)	Type of DLMs
Mazari et al. [125]	2019	MLGCN	98.6	GCN
Bianchi et al. [126]	2019	ArmaConv	96	GCN
WuFelix et al. [127]	2019	SGCConv	94	GCN

**Table 11 sensors-23-05121-t011:** The results of HAR on the KLHA3D-102 [100] and KLYOGA3D [14] datasets.

Datasets	Configurations	Methods				
		DDnet [120] *Acc* (%)	PA-ResGCN [59] *Acc* (%)	CNN-LSTM [128] *Acc* (%)	SgCNN [14] *Acc* (%)	CCNN [129] *Acc* (%)
KLHA3D-102	KLHA3D-102_Conf. 1	52.94	40.02	92.63 (Cross-Subject)	93.82	98.12 (Cross-Subject)
	KLHA3D-102_Conf. 2	45.18	52.94	92.46 (Cross-View)	-	96.15 (Cross-View)
	KLHA3D-102_Conf. 3	52.94	48.04	-	-	-
	KLHA3D-102_Conf. 4	2.55	10.22	-	-	-
	KLHA3D-102_Conf. 5	1.96	8.56	-	-	-
KLYoga3D	KLYOGA3D_Conf. 1	20.51	33.33	-	95.48	-
	KLYOGA3D_Conf. 2	25.64	53.85	-	-	-

**Table 12 sensors-23-05121-t012:** The processing time to recognize the human activity on the KLHA3D-102 [100] dataset.

Models	Processing Time (fps)
DDnet [120]	5000
PA-ResGCN [59]	50

## Data Availability

Not applicable.

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
