# Peer review of "Deep Learning for Human Activity Recognition on 3D Human Skeleton: Survey and Comparative Study"

_sensors, 2023, doi:10.3390/s23115121_

Round 1
Reviewer 1 Report
In this paper, the Authors are proposing a survey on using deep learning to recognize human activity based on three?dimensional (3D) human skeleton data as input.
They use a unified model from end-to-end for detecting, segmenting, estimating 3D human poses and recognizing human activities for training and learning exercises in Gym or Yoga for training and protecting health.
After carefully reading, I find that this paper is extremely interesting, however in order to further improve I would only recommend to improve the conclusions and more references on the background.
Author Response
Response to Reviewer 1

Reviewer 2 Report
Authors developed a deep learning networks for activity recognition based on extracted feature vectors: Recurrent Neural Network (RNN), Convolutional Neural Network (CNN), Graph Convolution Network (GCN) and Hybrid Deep Neural Network (Hybrid−DNN). The work seems to be promising and the paper is well written, but I have a unique suggestion for improving the paper: A detailed complexity analysis must be included in the paper versus the time consumed for each case.
Author Response
Response to Reviewer 2

Reviewer 3 Report
In this paper, the authors have surveyed methods of using deep learning to recognize human activities based on 3D human skeleton input data.The detailed comments are as follows. Please consider them for further revision.
1. On page 1, in the abstract, the article mentions that “Our research is based on three types of deep learning networks for activity recognition based on extracted feature vectors,” but later four types of networks are listed, which need to be corrected.
2. Figure 3 on page 4, Figure 6 on page 7, and Figure 10 on page 12 are not clear enough, so it is suggested that they be replaced with clearer images.
3. From pages 17 to 21, Tables 1 through 9 are suggested to be placed in section 3.4 corresponding to the table descriptions, with a detailed analysis of each table.
4. On page 22, it is mentioned that "This shows that the efficiency of learning features on the KLYOGA3D dataset is very low", what causes this phenomenon, please explain it. In addition, we can see from Figure 13 that when the accuracy of the test set is stabilized, the accuracy of the train set is not yet stabilized, do we need to continue training to make it stable? Please explain in the paper.
No
Author Response
Response to Reviewer 3

Reviewer 4 Report
The manuscript is relevant; however, there are some areas that need improvement in order to enhance the authors' contribution. For example, in the "Introduction" section, some information could be summarized or omitted to make it more concise. Specifically, the mention of HAR application fields could be shortened, focusing only on the most relevant ones for the study. The authors should provide a more detailed analysis of the challenges faced in recognizing activities based on 3D poses. This should include a discussion of the lack of sufficient information to distinguish actions with a limited number of reference points, as well as other specific challenges related to data dimensionality.
The authors should offer a more comprehensive analysis of the challenges encountered in activity recognition based on 3D poses, including a discussion of the insufficient information to distinguish actions with a limited number of reference points, as well as other specific challenges related to data dimensionality. It would be beneficial to include more information about the limitations and specific difficulties faced in this context.
While the manuscript mentions the results of activity recognition using different approaches, which is interesting, it lacks a critical analysis of the results or a comparison with related works. In reality, I would say that for the work conducted, the discussions, comparisons, and analyses could be improved.
Please review the formatting, tables, and figures.
Author Response
Response to Reviewer 4

Reviewer 5 Report
The author had selected a good topic for review on Deep Learning-based for Human Activity Recognition on 3D Human Skeletons. The author should make the following changes before acceptance:
1. The author should provide the authentication of Figure 1. From which source, the author takes this data because the author claimed that HAR recognition is based on Sensor, Vision, RFID and Device-free data. Justify it?
2. In Figure 8, the author shows the statistics of 4-year data. From where did the author consider this data, and Why only four years of data? Justify it.
3. In section 3.2: The 11 databases author has shown, but the author has conducted the 9 dataset studies in tables 1 to 9. What about the NTU RGB+D+120 database and KLYOGA 3D database literature work or literature table 10/11?
4. In section 3.3: the author has shown only three evaluation parameters for HAR recognition. Why not other parameters are shown in that section, like sensitivity, specificity, F-1 Score etc.? Justify it?
5. In section 3.4 heading is not suitable because this section shows the literature of all nine databases studied. How is it a result section?
6. What is the significance of Figure 13 (a, b, c, d)? Two we can understand for two different datasets? Describe it properly.
7. The conclusion section is an essential part of the manuscript, so the author must reframe the conclusion section concisely because this section needs to be appropriately elaborated.
8. In Section 3.5, the author claimed various challenges and discussions but nowhere discussed the challenges the researchers faced while implementing the HAR recognition models. It must be point-to-point with the pictorial view.
9. In Table 2: Reference numbers 53, 76 and 108 are repeated twice. Kindly rectify this.
10. In Table 8: Reference number 38 is repeated two times. Kindly rectify this.
11. All the references must be in the same format.
12. 97, no citation is missing in the text.
13. No grammatical or typos errors are there in the manuscript. Kindly rectify it.
14. Overall, the author must explain how systematically the author has done this study in the introduction section. How many papers they referred to and how they selected, and from where they referred.
Typos and grammatical errors are there.
Author Response
Response to Reviewer 5

Round 2
Reviewer 4 Report
Dear Authors
I am pleased to know that the recommended improvements have been implemented in the first version of the manuscript. However, I believe that there are still some points that require attention. For instance, I suggest making the section on HAR applications more concise, highlighting only the most relevant ones for the study. Additionally, a more detailed analysis of the challenges faced in recognizing activities based on three-dimensional poses would be beneficial. Nevertheless, I believe that these issues can be easily addressed through a quick revision and do not believe they impact the merit of the conducted research.
Author Response
Response to Reviewer 4
